# MicroRNA-7a2 Contributes to Estrogen Synthesis and Is Modulated by FSH via the JNK Signaling Pathway in Ovarian Granulosa Cells

**DOI:** 10.3390/ijms23158565

**Published:** 2022-08-02

**Authors:** Liuhui Li, Jinglin Zhang, Chenyang Lu, Bingjie Wang, Jiajia Guo, Haitong Zhang, Sheng Cui

**Affiliations:** 1College of Veterinary Medicine, Yangzhou University, Yangzhou 225009, China; dx120170116@yzu.edu.cn (L.L.); mx120170745@yzu.edu.cn (C.L.); wangbingjie123188@163.com (B.W.); 18562312703@163.com (J.G.); zhang_haitong@icloud.com (H.Z.); 2Institute of Reproduction and Metabolism, Yangzhou University, Yangzhou 225009, China; waldenl@126.com; 3Joint International Research Laboratory of Agriculture and Agri-Product Safety, The Ministry of Education of China, Yangzhou University, Yangzhou 225009, China; 4Jiangsu Co-Innovation Center for Prevention and Control of Important Animal Infectious Diseases and Zoonoses, Yangzhou 225009, China

**Keywords:** microRNA-7a2, estrogen synthesis, Glg1, Cyp19a1, JNK, ovary

## Abstract

MicroRNA-7a2 (miR-7a2) plays fundamental roles in the female reproductive axis, and estrogen is indispensable for maintaining ovary function. However, the interaction between miR-7a2 and ovarian function is unclear. The present study aimed to determine whether and how miR-7a2 functions in estrogen synthesis. Firstly, the results verified that miR-7a was highly expressed in ovarian granulosa cells. The knockout (KO) of miR-7a2 caused infertility and abnormal ovarian function in mice. Concomitantly, the *C**yp19a1* expression and estrogen synthesis were significantly inhibited, which was validated in primary granulosa cells. The mice transplanted with miR-7a2 KO ovaries showed similar results; however, estrogen supplementation reversed infertility. In the in vitro experiment, follicle-stimulating hormone (FSH) significantly improved the expression of miR-7a and *C**yp19a1* and the synthesis of estrogen. However, the miR-7a2 KO markedly reversed the function of FSH. Also, FSH upregulated miR-7a by activating the (c-Jun N-terminal kinase) JNK signaling pathway. In addition, Golgi apparatus protein 1 (*Glg1*) was shown to be the target gene of miR-7a2. These findings indicated that miR-7a2 is essential for ovarian functions with respect to estrogen synthesis through the targeted inhibition of the expression of *G**lg1* and then promoting *C**yp19a1* expression; the physiological process was positively regulated by FSH via the JNK signaling pathway in granulosa cells.

## 1. Introduction

Reproduction is regulated by various complex mechanisms, such as inheritance, endocrine, and metabolism, which have gained significant interest. The hypothalamus-pituitary-ovary (HPO) axis plays a fundamental role in female neuroendocrinology and reproduction [1]. In females, the luteinizing hormone (LH) triggers ovulation and luteinization, whereas the follicle-stimulating hormone (FSH) stimulates follicle maturation and estrogen secretion [2,3]. In turn, estrogen from the ovary also regulates these gonadotropins through a negative feedback loop [4]. Notably, estrogen plays a critical role in the reproduction process. Also, it is the primary determinant of pituitary neuron function, facilitating the cells to exhibit fluctuating patterns of biosynthesis and secretion activity [5]. The interactions between the ovary and the hypothalamus-pituitary are designed to promote the generation of a naturally fertilizable ovum [6], and the ovary provides essential structural support for the development of the ovum throughout evolution [7]. Moreover, estrogen secreted by the ovary is obligatory for normal folliculogenesis beyond the antral stage [8]. Female mice lacking the capacity to produce estrogen have arrested folliculogenesis and are infertile [9]. These pieces of evidence confirm the importance of estrogen in maintaining normal reproductive function.

In the ovary, estrogen is synthesized by the synergistic action of granulosa cells and theca cells of the follicles [10]. Typically, granulosa cells express cytochrome P450 family 19 subfamily A member 1 (Cyp19a1) and 17β-hydroxysteroid dehydrogenase (17β-HSD), while theca cells express Cyp17a1 [7]. Using cytochrome P450 17A1 (the production of the *Cyp17a1* gene), theca cells produce androstenedione, which diffuses to granulosa cells and is converted to estrone by P450 aromatase (the production of the *C**yp19a1* gene) and then into estradiol by 17-HSD [7,11]. Accumulating evidence demonstrated that microRNAs (miRNAs) are involved in ovarian biological processes. As reported, the miRNA-200s (miR-200s) on chromosome 23 plays an important role in controlling oocyte maturation by regulation of the LH expression [12]. In addition, Lynda K. has reviewed the modulation effects of miRNAs on the biology of the ovary, including its follicular development, oocyte maturation, and ovulation [13]. In particular, some miRNAs, such as miR-378 and miR-146b, have demonstrated modulating estrogen synthesis by regulating aromatase [14,15]. However, little is known about the functional involvement of miRNAs in the ovary and estrogen synthesis.

MiRNA-7 (miR-7) is evolutionarily conserved across many mammals [16,17] and harbors a conserved seed sequence derived from miR-7a1, miR-7a2, and miR-7b precursors; among these, miR-7a1 and miR-7a2 encode the same sequence of miR-7a [18]. Previous studies have demonstrated that miR-7a2 is involved in tissue development, hormone synthesis, and secretion. For example, miR-7a2 promotes pituitary stem cell development [19], prolactin production [20], and insulin secretion [21]. Ahmed et al. reported that miR-7a2 knockout (KO) mice induce hypogonadism by disrupting the FSH/LH synthesis and secretion, while genetic deficiency miR-7a1 or miR-7b exerts little impact [18]. However, whether miR-7a2 is expressed in ovaries and affects the physiological function of the ovaries is yet unknown. The present study aimed to investigate the effects of miR-7a2 on ovary functions and the underlying mechanisms.

## 2. Results

### 2.1. miR-7a Is Expressed in Mouse Ovaries

To determine the function of miR-7a in mouse ovaries, the expression of the miRNA was examined in adult mouse ovaries. ISH results showed the localization of miR-7a in the ovarian follicles, especially granulosa cells (Figure 1A,B). Then, the *ISH* and immunofluorescence triple staining results demonstrated that miR-7a were co-located with FSHR (a marker of the granulosa cell) in the same cells but not with DDX4 (a marker of oocyte). These findings suggested that miR-7a is highly expressed in ovarian granulosa cells (Figure 1C).

### 2.2. miR-7a2 KO Impairs Ovarian Function

MiR-7a2 systemic KO mice were used to investigate the effects of miR-7a2 on ovarian function. The expression of miR-7a was 64.8% lower in KO mice ovaries than in WT mice (Figure 2A). In addition, WT mice had a normal estrus cycle, whereas the miR-7a2 KO mice remained in the diestrus stage (Figure 2B). Furthermore, miR-7a2 KO mice were infertile (Figure 2C), and their ovaries were notably smaller than those of the WT mice (Figure 2D). The histological analysis revealed that there were no mature follicles and corpus luteum in the miR-7a2 KO mouse ovaries, and the number of primordial follicles and atretic follicles was significantly increased (Figure 2E,F). As shown in Figure 2G–I, the *C**yp19a1* mRNA level in the miR-7a2 KO mice was decreased by 78.0%, whereas *Cyp17a1* and *17β-HSD* mRNA levels were not affected. Furthermore, the level of serum E_2_ was decreased by 56.0% in the miR-7a2 KO mice (Figure 2J). These results suggested that miR-7a2 deletion impairs ovarian function.

### 2.3. Deficiency of miR-7a2 Impairs the Function of Granulosa Cells in Estrogen Synthesis

Granulosa cells are involved in estrogen production in the ovaries. Therefore, primary granulosa cells were collected from the dominant follicles of the WT and KO mice to examine the effect of miR-7a2 on estrogen synthesis. The expression levels of miR-7a, E_2_, *C**yp19a1*, and *17β-HSD* were determined in primary granulosa cells in vitro. As shown in Figure 3A, the miR-7a level in granulosa cells from the miR-7a2 KO mice was decreased by 70.0% compared to the level in the WT mice. In addition, the *C**yp19a1* mRNA level in the granulosa cells from the miR-7a2 KO mice was decreased by 87.5% (Figure 3B). However, the *17β-HSD* mRNA expression did not show a marked difference (Figure 3C). RIA demonstrated that the concentration of E_2_ in the medium was 75% lower in the KO group than in the control group (Figure 3D), which was consistent with the in vivo results.

### 2.4. Specific Deficiency of Ovarian miR-7a2 Impairs the Ovarian Function

To investigate whether miR-7a2 directly affects ovarian function, “WT + KO Ovary” mice and “WT + WT Ovary” mice were used. The estrus cycle of these mice was examined by a vaginal smear 21 days after organ transplantation. As shown in Figure 4A, “WT + WT Ovary” mice had a normal estrus cycle; however, “WT + KO Ovary” mice had a prolonged diestrus stage accompanied by a significantly shortened estrus. Furthermore, compared to the “WT + WT Ovary” mice, “WT + KO Ovary” mice were infertile (Figure 4B), and their ovaries were notably smaller (Figure 4C). Histological examination showed a lack of corpus luteum in “WT + KO Ovary” mice (Figure 4D). Moreover, the *C**yp19a1* mRNA expression in the ovaries of “WT + KO Ovary” mice was decreased by 99% compared to that in the control mice, whereas the *Cyp17a1* and *17β-HSD* mRNA expressions did not show any significant difference (Figure 4E–G). Furthermore, the E_2_ level of “WT + KO” mice decreased by 45% compared to that of “WT + WT” mice (Figure 4H). These results demonstrated that miR-7a2 played a major role in ovarian function.

### 2.5. Estrogen Reverses the Infertility of Female Mice Caused by Ovarian miR-7a2 Deletion

“WT + KO Ovary” mice were administered E_2_ to assess whether estrogen could rescue infertility induced by the miR-7a2 KO. Compared to mice without the E_2_ supplement, those in the rescue group had a longer estrus period and shorter diestrus period (Figure 5A). The litter size of the “WT + KO Ovary + E_2_” group was 4.25. On the other hand, the litter size of the “WT + KO Ovary” group was zero (Figure 5B). These results indicated that estrogen reverses the infertility of female mice caused by miR-7a2 deficiency. Moreover, the E_2_ level of mice serum in the “WT + KO Ovary + E_2_“ group was upregulated by 46.7% compared to the “WT + KO Ovary” group (Figure 5C).

### 2.6. FSH Regulates Estrogen Synthesis by Modulating miR-7a in Granulosa Cells

Primary granulosa cells were treated with FSH (200 ng/mL) for 6 h, and the levels of *miR-7a* and *C**yp19a1* mRNA in the cultured cells and E_2_ in the cell media were determined. The results showed that the FSH treatment increased *miR-7a* and *C**yp19a1* mRNA and the E_2_ levels by 88.5%, 59.7%, and 61.9%, respectively, in WT granulosa cells. However, the FSH treatment had no significant effect on the miR-7a2 KO primary granulosa cells (Figure 6A–C). These results suggested that FSH upregulates miR-7a, which might be essential for the regulatory effect of FSH on estrogen.

### 2.7. FSH Enhances the Expression of miR-7a via JNK Signaling Pathway in Granulosa Cells

In order to identify the mechanism of FSH affecting miR-7a and estrogen synthesis, primary granulosa cells were incubated with SP [SP 600125, c-Jun N-terminal kinase (JNK) inhibitor, 10 μM], CH [protein kinase C (PKC) inhibitor, 1 μM], H-89 [protein lysine acetyltransferase (PKA) inhibitor, 10 μM], SB [SB 203580, p38 inhibitor, 10 μM], and PD [PD 98059, extracellular regulated MAP kinase 1/2 (ERK1/2) inhibitor, 10 μM] separately for 1 h, followed by an FSH (200 ng/mL) treatment for 6 h. The results showed that the JNK inhibitor abrogated the effect of FSH on miR-7a, whereas other signaling pathway inhibitors had no significant effect (Figure 7A). In addition, the expression of p-JNK was examined by Western blotting, and the results showed that p-JNK expression was inhibited by SP (Figure 7B). To determine whether JNK is involved in the regulation of FSH on the *C**yp**19a1* expression and estrogen synthesis, the granulosa cells were treated with SP or SP combined with FSH. Subsequently, the *C**yp**19a1* expression and estrogen concentration were estimated in the culture media. The results showed that FSH significantly increased the expression of *C**yp**19a1* mRNA (Figure 7C) and the E_2_ concentration, inhibited by the SP treatment (Figure 7D), deeming miR-7a as a critical downstream molecule of the JNK signaling pathway.

### 2.8. Glg1Is the Target Gene of miR-7a in Ovarian Granulosa Cells

The downstream target genes of miR-7a were searched using the related microRNA analysis programs (miRBase; http://www.microrna.org, accessed on 14 December 2021 and TargetScan; http://www.targetscan.org/, accessed on 14 December 2021). Four genes—*P**ki**α* (protein kinase inhibitor, alpha), *S**p**1* (trans-acting transcription factor 1), *C**dc**25**b* (cell division cycle 25B), and *G**lg**1* (Golgi apparatus protein 1)—were considered as the putative miR-7a target genes. Then, the mRNA levels of the four target genes were detected after 12 h post-transfection with 50 nM miR-7a mimics (miR-7a-mi). The results showed that miR-7a upregulation did not have significant effects on Pkiα, Sp1, and Cdc25b but significantly decreased *G**lg**1* mRNA by 53.8% (Figure 8A). Subsequently, the RNAHybrid database (http://bibiserv.cebitec.uni-bielefeld.de/rna accessed on 23 December 2021) provided miR-7a binding site predictions by offering the minimum free energy of hybridization between the 3′-untranslated region (UTR) of *G**lg**1* and the seed sequence of miR-7; the corresponding minimum free energy of *G**lg**1* was –25.8 kcal/moL (Figure 8B,C). Then, the psiCHECK^TM^-2.0 vector was used to clone the putative 3′-UTR target site downstream of a luciferase reporter gene (Figure 8D). The dual-luciferase reporter assay demonstrated that the luciferase activity did not have significant effects on 293FT cells transfected with miR-7a-mi. On the other hand, the luciferase activity was suppressed by 30.9% in the cells by co-transfecting with miR-7a-mi and *G**lg**1*-3′UTR-psiCHECK^TM^2.0 compared to the cells transfected with *G**lg**1*-3′UTR-psi CHECK^TM^2.0 (Figure 8E).

The RT-qPCR results showed that miR-7a had a significant inhibitory effect on the *G**lg**1* mRNA levels in both the ovary and granulosa cells (Figure 8F). In addition, the results of Figure 8G confirmed that the expression of *G**lg**1* was negatively correlated with the expression of miR-7a. Finally, to detect whether *G**lg**1* acts as a signaling molecule in the FSH-regulated E_2_ synthesis signaling pathway, 200 nM nc-siRNA or Glg1-siRNA was transfected in the primary granulosa cells for 6 h, followed by another 6 h incubation with or without FSH. The results showed that Glg1-siRNA significantly increased the *C**yp**19a1* mRNA levels and E_2_ concentrations by 89.69% and 32.36%, respectively, compared to the nc-siRNA. In nc-siRNA transported cells, the FSH treatment increased the *C**yp**19a1* mRNA expression and E_2_ concentration by 66.88% and 104.01%, respectively. However, this effect of FSH was abolished in the cells transfected with Glg1-siRNA (Figure 8H,I). The results indicated that *G**lg**1* is the target gene of miR-7a in ovarian granulosa cells.

## 3. Discussion

The function of miR-7a2 in various endocrine organs has been reported in several studies [18,20,22,23]. Previous studies demonstrated that miR-7a2 affects the gonad development and reproductive function of mice by modulating the synthesis and secretion of the FSH and LH. Herein, we demonstrated a crucial role of miR-7a2 in ovarian function regulation via a non-gonadal axis pathway. The study provides a new perspective for understanding miR-7a in the maintenance of normal ovarian function and estrogen synthesis.

In the present study, miR-7a2 KO affected the estrus cycle and litter rate of female mice, characterized by persistent diestrus period and infertility, as described previously [18]. Furthermore, hematoxylin-eosin (HE) staining indicated that the ovaries of the miR-7a2 KO mice lacked mature follicles and corpus luteum, which were replaced by immature and atretic follicles. These results indicated that an miR-7a2 deficiency affects the normal physiological function of the ovaries. Previous studies have reported that ovarian granulosa cells are involved in follicular growth, development, and atresia [24,25,26], and the proliferation of granulosa cells is essential for the normal function of the ovaries [26]. The main function of granulosa cells is the synthesis of estrogen, a critical hormone for follicular development regulated by several enzymes, including *Cyp17a1*, *C**yp19a1*, and *17β-HSD* [27]. Thus, the synthesis of estrogen and these enzymes were evaluated. The results revealed that miR-7a2 KO significantly decreased the *C**yp19a1* expression and estrogen synthesis. However, the expression of *Cyp17a1* and *17β-HSD* was not affected. Therefore, it could be inferred that miR-7a2 participates in ovarian function by modulating estrogen synthesis via the *C**yp19a1* expression regulation. MiR-7a2 systemic KO mice exhibit the abnormal function of the pituitary associated with fertility [18]. Therefore, a model of ovarian transplantation was used to confirm that the specific loss of miR-7a2 in the ovaries reduced estrogen synthesis, which contributed to low fertility. Inevitably, the decrease in estrogen levels reduces uterine receptivity. Therefore, an estrogen supplementation model was used to evaluate fertility. In the present study, the litter size was increased, and infertility was rescued. The findings indicated the importance of estrogen for the fertility of female mice and emphasized estrogen regulation by miR-7a2 in the maintenance of ovarian function. The expression *C**yp19a1* of ovaries in “WT + KO Ovary + E_2_” mice was significantly increased compared to the control ovaries. As granulosa cells are the primary sites of estrogen production [28], primary granulosa cells were used to determine whether miR-7a2 affects estrogen synthesis by regulating *C**yp19a1*. The results showed that the expression levels of miR-7a2 and *C**yp19a1* and estrogen synthesis in granulosa cells were significantly lower in the miR-7a2KO mice than in the WT mice, indicating that these primary cells may be used to investigate estrogen synthesis. Since *C**yp19a1* is a key enzyme for estrogen synthesis, miR-7a2 may modulate the secretion of estrogen by regulating the expression of *C**yp19a1* in granulosa cells. Nonetheless, how miR-7a2 affects the expression of *C**yp19a1* and estrogen synthesis is yet to be elucidated.

MicroRNAs bind to the 3′-UTR complementary elements of target mRNAs to inhibit target mRNA translation or promote gene silencing by degrading the mRNAs [29,30]. In this study, miR-7a upregulated the expression of *C**yp19a1*, indicating that *C**yp19a1* may not be the direct target gene of miR-7a. The dual-luciferase reporter assay demonstrated that *G**lg1* is the target gene of miR-7a, which inhibits the expression of *C**yp19a1* and estrogen synthesis. Although there is no study linking *G**lg1* to steroidogenesis, *G**lg1* has been shown to inhibit FSH and LH synthesis and secretion [18,22] by suppressing bone morphogenetic protein 4 (BMP4) signaling [18]. In the ovary, BMP4 is shown to promote the expression of *Cyp19a1* and estrogen production in mouse ovarian granulosa cells [31]. Thus, BMP4 can link the effects of miR-7a-Glg1 on estrogen synthesis. However, how miR-7a-Glg1 affects this process needs to be investigated further.

The FSH is vital for estrogen synthesis [32]. Herein, we treated the cells with the FSH and found that the hormone significantly increased *Cyp19a1* and estrogen in normal cells. However, these effects were not observed in the KO cells. Notably, the increase in the FSH also caused a significant increase in the expression of miR-7a2 in the WT cells, indicating that miR-7a2 acts as a downstream regulator of the FSH, playing a critical role in regulating the *Cyp19a1* expression and estrogen synthesis through the non-FSH pathways. Some studies have reported that the FSH promotes estrogen synthesis and the generation of mature follicles by regulating a related gene expression in target granulosa cells, including the PKC [33], PKA [34], ERK [33], p38 [34], and JNK [35] signaling pathways. Therefore, SP600125 (JNK), H-89 (PKA), SB203580 (p38), PD98059 (ERK), and CH (PKC) were used as the inhibitors of these signaling pathways. Our results demonstrated that miR-7a2 regulates estrogen synthesis in ovarian granulosa cells, which was further controlled by the FSH via the JNK signaling pathway. This finding provides a theoretical basis for understanding miR-7a2-regulated estrogen synthesis.

In summary, miR-7a is highly expressed in mouse ovarian granulosa cells and affects ovarian function by inhibiting *G**lg1* and indirectly modulating the *Cyp19a1* expression and estrogen synthesis. Itis regulated by the FSH via the JNK signaling pathway. These findings guide the regulation of estrogen synthesis to modulate ovarian function.

## 4. Materials and Methods

### 4.1. Animals

The miR-7a2 systemic KO mouse strain was established using the CRISPR/Cas9 system [19] in adult ICR mice purchased from the Yangzhou University Comparative Medical Center. All mice were raised under controlled conditions (22–25 °C, 12 h light/dark cycle, 50 ± 10% humidity) with free access to food and water. The animal studies (including the mice euthanasia procedure) were conducted in compliance with the Yangzhou University institutional animal care regulations and the AAALAC and IACUC guidelines.

The ovarian transplantation mouse model was established as described previously [36,37]. Briefly, the ovaries of wild-type (WT) or miR-7a2 systemic KO mice, aged 21 days, were transplanted into the ovarian bursa of WT mice. The ovarian transplanted mice were divided into the “WT+WT Ovary” group, the “WT + KO Ovary” group, and the “WT + KO Ovary + estradiol (E_2_)” group. In the “WT + WT Ovary” group, the ovaries of WT mice were transplanted into the ovarian bursa of WT mice on both sides. In the “WT + KO Ovary” group, the ovaries of KO mice were transplanted into WT mice on both sides. In the “WT + KO Ovary + E_2_” group, the ovaries of WT mice were transplanted into WT mice on one side of the ovarian bursa with ligation of the oviduct on the same side, and the ovaries of the KO mice were transplanted on the other side. Specifically, an incision on the opposite of the fallopian tube was made in the ovarian bursal membrane using forceps to remove the ovary. Then, the ovary of the donor mouse was immediately transplanted into the ovarian bursa of the recipient mice. After a 3-week recovery period, further experiments were performed with the transplantation mice. For fertility assessment, female mice were separately mated with WT male mice for 3 weeks.

### 4.2. In Situhybridization (ISH) and Immunofluorescence Triple Staining

The ovaries of 6–8-week-old mice were collected, fixed in 4% paraformaldehyde (PFA), incubated at 4 °C for 2 h, and immersed in 30% sucrose solution until the tissues sank. Then, the ovaries were embedded with an embedding medium to ensure optimal cutting temperature and sliced into 10-μm-thick sections using a cryostat. The sections were treated with acetylation solution, followed by 0.1% Triton for 10 min. Then, the tissue sections were covered by an appropriate amount of pre-hybrid solution and incubated in a wet box at room temperature for 6 h. Subsequently, the slices were incubated with a hybridization solution containing the target or scrambled probe (for the negative control) (1 pmoL/150 μL; Exiqon, Woburn, MA, USA), respectively, at 47 °C overnight. The sections were washed twice with a 0.2× saline-sodium citrate buffer (SSC) for 1 h and incubated with a blocking buffer at room temperature for 1 h before probing with alkaline phosphatase (AP)-conjugated antibody against DIG (1:1000; Roche Diagnostics, Basel, Switzerland) at 4 °C overnight. After two washes with the AP buffer, the sections were incubated with nitrotetrazolium blue chloride and 5-bromo-4-chloro-3-indolylphosphate p-toluidine salt in the dark to develop the staining. The reaction was terminated using distilled water, and the slides were observed and photographed under a Leica DM750 microscope (Leica, Wetzlar, Germany).

After the ISH procedure above, the slides were treated with 10% BSA in PBS and incubated with 1:50 anti-FSHR (ab113421, Abcam, Cambridge, MA, USA) and 1:100 DDX4 (abcam, ab27591) at 4 °C overnight. After washing, the sections were subsequently incubated with the secondary antibodies, which were 1:200 cy2-conjuncted donkey anti-rabbit IgG (711-225-152, Jackson ImmunoResearch, West Grove, PA, USA) and cy3-conjuncted goat anti-mouse IgG (Jackson ImmunoResearch, 115-165-003) for 3 h and kept in darkness at room temperature. The nuclei were stained by DAPI (Roche Applied Science, 10236276001) for 30 min. Then, the sections were observed and photographed under a Leica DM750 microscope (Leica, Wetzlar, Germany).

### 4.3. Vaginal Smear

The estrus cycle was monitored by examining the vaginal smears obtained between 8:00 a.m. and 11:00 a.m. for 18 consecutive days before measuring ovarian functionality. The smears containing vaginal cells were stained with Wright’s stain. The different stages of the estrus cycle were determined according to the predominant cell type present in the vaginal smears [38].

### 4.4. Radioimmunoassay (RIA)

The concentrations of E_2_ in the serum and medium were measured using RIA reagents (Beijing North Institute Biological Technology, Beijing, China) according to the manufacturer’s instructions. For each RIA, the intra- and inter-assay coefficients of variation were <10%.

### 4.5. Ovarian Granulosa Cell Culture and Treatment

Each of the miR-7a2 KO and WT mice were intraperitoneally injected with 5 IU pregnant mare serum gonadotropin (PMSG) on post-natal day 21. After 44–46 h, the mice were sacrificed by cervical dislocation. Bilateral ovaries were removed under sterile conditions. The adipose tissue around the ovaries was peeled off under a stereomicroscope (Olympus, Tokyo, Japan). The cumulus-oocyte complex and granulosa cells were released using a 1 mL syringe needle by puncturing the dominant follicles (>200 μm in diameter) on the ovaries [39,40]. The FSH receptor was used to identify the purity of the ovarian granulosa cells [41]. After the oocytes were aspirated, the granulosa cells were filtered through a sieve with a pore size of 0.074 mm and seeded into a 6-well plate. Subsequently, the cells were cultured in Dulbecco’s modified Eagle’s medium/F12 (DMEM/F12) containing 2 mM glutamine and 10% fetal bovine serum (FBS) in an incubator at 37 °C with 5% CO_2_. The media were refreshed every 48 h. At 80–90% confluence, the primary granulosa cells were cultured for 12 h with a serum-free medium and stimulated with the FSH (200 ng/mL) for 6 h. Subsequently, the culture medium was collected to measure the level of estrogen by RIA.

### 4.6. Real-Time Quantitative Polymerase Chain Reaction (RT-qPCR)

The total RNA was extracted from the granulosa cells using the RNAiso Plus (SD1412; TaKaRa, Kyoto, Japan) method, according to the manufacturer’s instructions. Reverse transcription (RT) was performed using the M-MLV Reverse Transcriptase kit (Promega, Madison, WI, USA), according to the manufacturer’s protocol. For mRNA RT, the reactions included purified RNA (500 μg/μL) and 50 nM RT stem-loop primers. The gene expression levels were measured using SYBR Premix Ex Taq (Q311-02; Vazyme; Nanjing, China) on an ABI 7500 Real-Time PCR System (Applied Biosystems; Foster, CA, USA). The level of *miR-7a* was normalized to the endogenous expression level of *U6*, while the levels of other mRNAs were normalized to the endogenous expression level of *GAPDH*. The primer sequences are listed in Table 1.

### 4.7. Western Blot

After stimulation, the primary granulosa cells were lysed in an ice-cold RIPA buffer (APPLYGEN; Beijing, China) supplemented with protease and phosphatase inhibitors (APPLYGEN). After the protein quantification using the BCA method, an equivalent of 30 μg of protein was separated by 12% SDS-PAGE and transferred to polyvinylidene difluoride (PVDF) membranes. The membranes were blocked with 5% skim milk at room temperature for 1 h and probed with the following primary antibodies at 4°C overnight: JNK (9258), p-JNK (4668), and GAPDH (5174) (Cell Signaling Technology; Danvers, MA, USA). All the antibodies were diluted at 1:1000. After washing with 0.1% Tween-20 in Tris-buffered saline (TBST), the membranes were incubated with the anti-goat IgG horseradish peroxidase (HRP) secondary antibody (111-035-003) (Jackson Immuno Research; West Grove, PA, USA) at room temperature for 1 h. Finally, the immune reactive bands were detected using an enhanced chemiluminescence (ECL) kit on the ChemiDoc XRS system (Bio-Rad, Marnes-la-Coquette, France).

### 4.8. Dual-Luciferase Reporter Assay

The dual-luciferase reporter genes were constructed using the psiCHECK^TM^-2.0 luciferase reporter vector. The vector containing *Not I* and *Xho I* sites and the miR-7a binding sites of *G**lg1* and 5′-GCUUGU-3′ was used for the reporter assay. The 293FT cells were cultured in 10% FBS-DMEM/F12 in 24-well plates and transfected using a Lipofectamine 2000 agent (Invitrogen Life Technologies, Gaithersburg, MD, USA) with a mixture containing the dual-luciferase reporter plasmid, and negative control of miR-7a mimics (nc-mi) or miR-7a mimics (miR-7a-mi) for 24 h, as described previously [22,23]. The luciferase activity was analyzed using the dual-luciferase reporter assay kit (E1910; Promega). Data were presented as fold-activity over Renilla luciferase activity. All transfection experiments were performed at least three times independently.

### 4.9. Statistical Analyses

All data were expressed as the mean ± standard error of the mean (SEM) of at least three independent experiments and evaluated using a one-way analysis of variance (ANOVA), followed by a Student’s *t*-test between different groups with SPSS software version 22 (22.0.0.0, IBM Corp, New York, NY, USA). *p* < 0.05 indicated statistical significance.

## Figures and Tables

**Figure 1 ijms-23-08565-f001:**
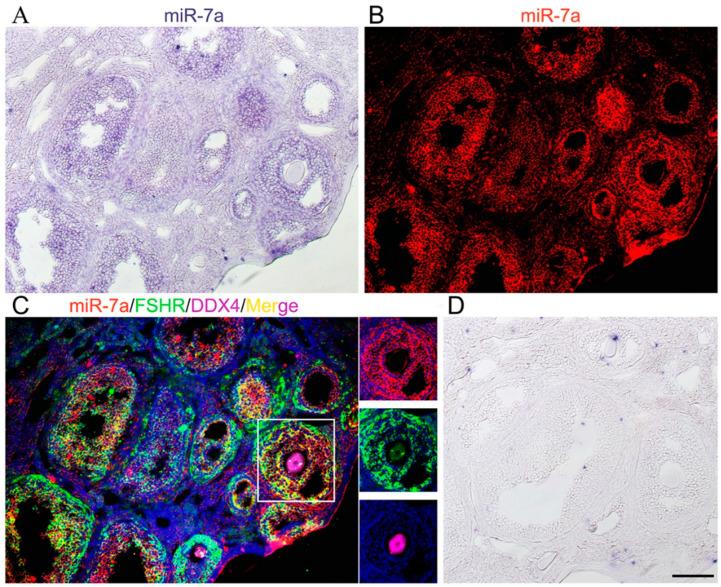
Expression of miR-7a in the mouse ovary.(**A**) The expression of miR-7a in the ovaries of mice at 6–8 weeks of age was detected by the ISH method. (**B**) miR-7a was adjusted from purple/ blue to red fluorescence. (**C**)Triple -immunofluorescence of adult mice ovary sections: miR-7a (red), FSHR (green), DDX4 (pink), and DAPI (blue); yellow is the merge of red and green. (**D**) The negative control of (**A**). Bars: 50 μm.

**Figure 2 ijms-23-08565-f002:**
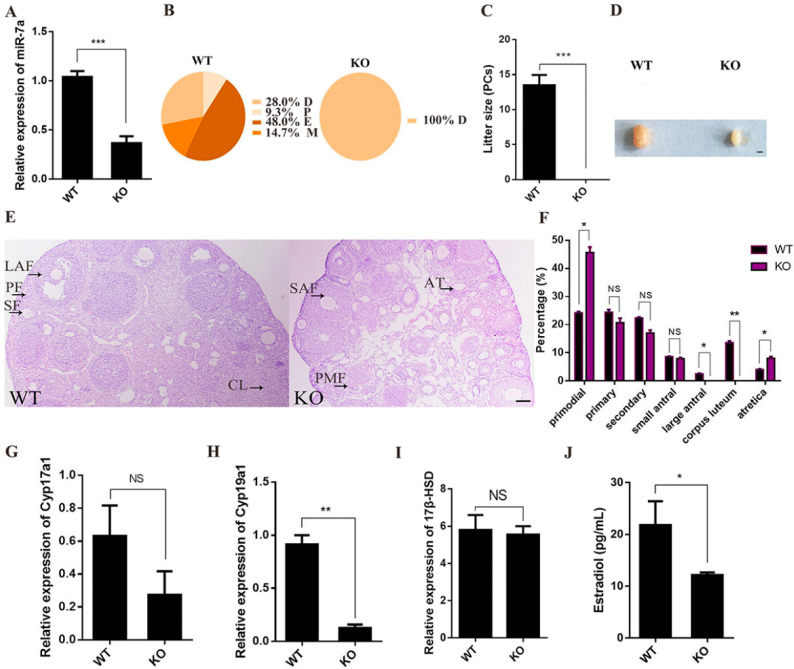
MiR-7a2 KO affects ovarian function. (**A**) KO efficiency of miR-7a2 in the ovaries of adult KO mice. (**B**) The estrus cycle of mice was determined by a vaginal smear for 18 consecutive days, comprising dioestrus, proestrus, estrus, and metestrus represented by D, P, E, and M, respectively; *n* = 6. (**C**) miR-7a2 KO mice and WT mice were separately mated with WT male mice for 3 weeks, and the litter sizes were recorded; *n* = 6. (**D**) Macro-observation of ovaries from adult WT and miR-7a2 KO mice; bars: 1000 μm. (**E**) Ovary sections of adult WT and miR-7a2 KO mice were observed by H.E. staining. PMF: primodial follicle; PF: primary follicle; SF: secondary follicle; SAF: small antral follicle; LAF: large antral follicle; CL: corpus luteum; AT: atretica. Bars: 100 μm. (**F**) The number of different follicles in the ovaries of WT and miR-7a2KO mice were analyzed via serial sections; *n* = 4. (**G**–**I**) The expression of *C**yp19a1*, *Cyp17a1*, and *17β-HSD* at the mRNA level in ovaries of adult WT and miR-7a2 KO mice was tested by RT-qPCR; *n* = 4. (**J**) E_2_ levels in the serum of adult WT and miR-7a2 KO mice were detected by RIA. Data are presented as mean ± SEM. * *p* < 0.05, ** *p* < 0.01. *****
*p* < 0.001. NS, not significant.

**Figure 3 ijms-23-08565-f003:**
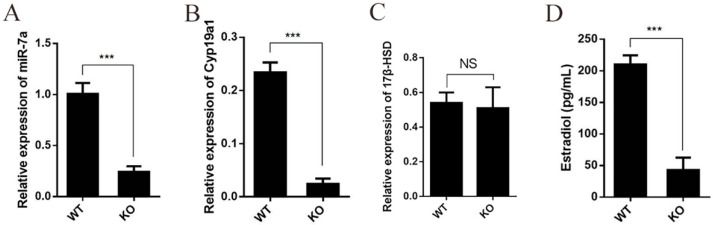
MiR-7a2 deficiency reduces estrogen synthesis in vitro. (**A**) The KO efficiency of miR-7a2 in primary granulosa cells. (**B**) The expression of *C**yp19a1* in primary granulosa cells of WT and KO mice. (**C**) The expression of *17β-HSD* in primary granulosa cells of WT and KO mice; *n* = 4. (**D**) Concentrations of E_2_ in culture medium. *** *p* < 0.001. NS, not significant.

**Figure 4 ijms-23-08565-f004:**
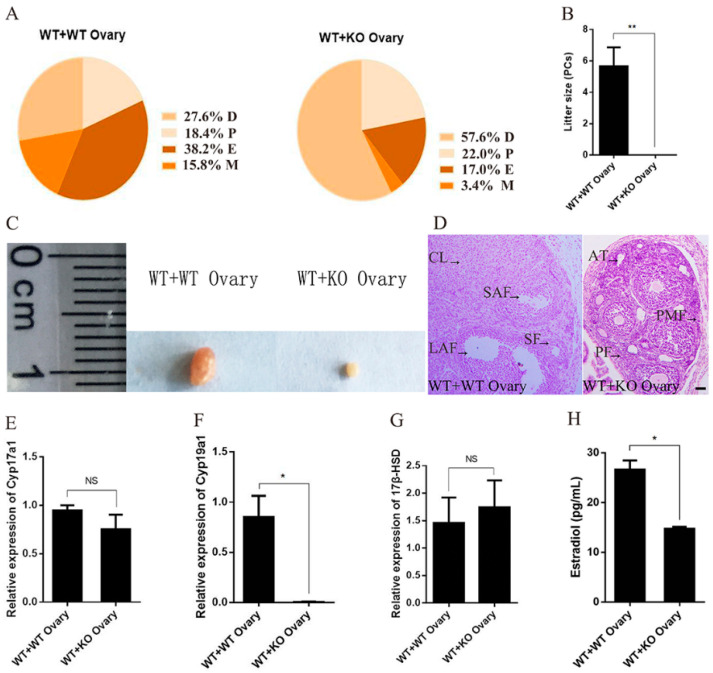
Deficiency of miR-7a2 in the ovary impairs ovarian function. (**A**) The estrus cycle of transplanted mice was determined by a vaginal smear for 18 consecutive days, comprising dioestrus, proestrus, estrus, and metestrus represented by D, P, E, and M, respectively; *n* = 6. (**B**) Ovary-transplanted mice were separately mated with WT male mice for three weeks, and the litter sizes were recorded; *n* = 6. (**C**) Macro-observation of ovaries from “WT + WT Ovary” mice and “WT + KO Ovary” mice. (**D**) Sections of ovary-transplanted mice were observed by H.E. staining; PMF: primodial follicle; PF: primary follicle; SF: secondary follicle; SAF: small antral follicle; LAF: large antral follicle; CL: corpus luteum; AT: atretica. Bars: 50 μm. (**E**–**G**) RT-qPCR was used to detect the expression of synthase *Cyp17a1*, *C**yp19a1*, and *17β-HSD* at the mRNA level in the ovaries of ovarian transplanted mice. (**H**) Detection of estrogen levels in the serum of ovarian transplanted mice. Data are presented as mean ± SEM. * *p* < 0.05. ** *p* < 0.01. NS, not significant.

**Figure 5 ijms-23-08565-f005:**
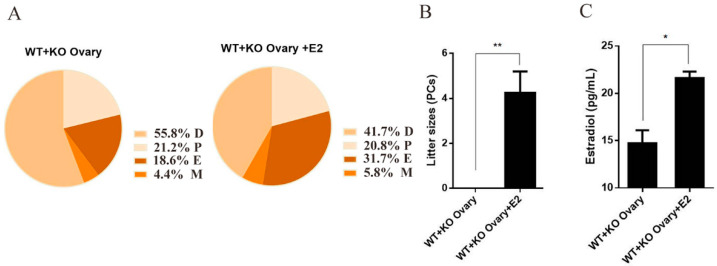
Estrogen reverses the infertility of female mice caused by miR-7a2 deletion. (**A**) The estrus cycle of transplanted model mice was determined by a vaginal smear for 18 consecutive days, comprising the diestrus, proestrus, estrus, and metestrus stages represented by D, P, E, and M, respectively (*n* = 4). (**B**) “WT + KO Ovary” mice and “WT + KO Ovary + E_2_“ mice were separately mated with WT male mice for 3 weeks, and the litter sizes were recorded (*n* = 4). (**C**) Detection of estrogen levels in the serum of ovarian transplanted mice (*n* = 4). Data are presented as mean ± SEM. * *p* < 0.05. ** *p* < 0.01.

**Figure 6 ijms-23-08565-f006:**
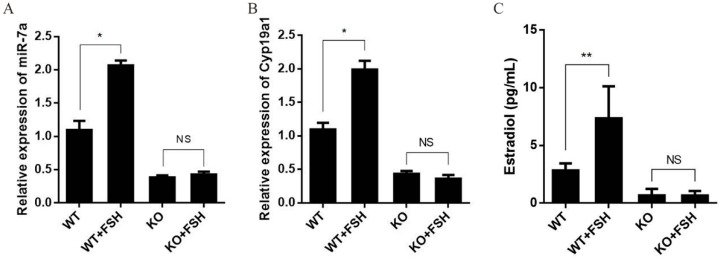
MiR-7a regulates the estrogen synthesis by FSH. (**A**) FSH increases the relative expression of miR-7a2 in the primary granulosa cells of WT mice. (**B**) Relative expression of *C**yp19a1* in mouse primary granulosa cells. (**C**) Effect of FSH on estrogen levels in the culture medium. * *p* < 0.05, ** *p* < 0.01. NS, not significant.

**Figure 7 ijms-23-08565-f007:**
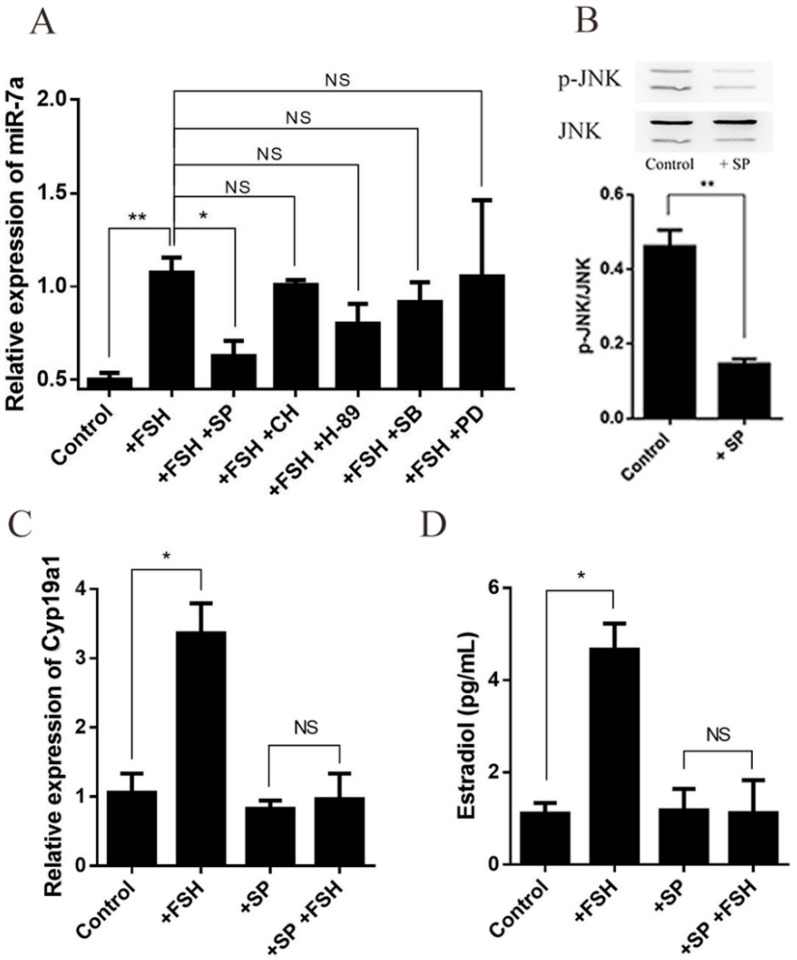
MiR-7a is regulated by FSH via the JNK signaling pathway in mouse granulosa cells. (**A**) MiR-7a levels in granulosa cells were determined by RT-qPCR. (**B**) Analysis of p-JNK protein in primary granulosa cells with SP (JNK inhibitor, 10 μM). (**C**) Quantification of the *C**yp**19a1* expression in cells by RT-qPCR (normalized to *GAPDH*). (**D**) Estimation of estrogen concentration in the cell culture medium by RIA. Data are presented as mean ± SEM (*n* = 3). Significant differences are indicated by * *p* < 0.05, ** *p* < 0.01. NS, not significant.

**Figure 8 ijms-23-08565-f008:**
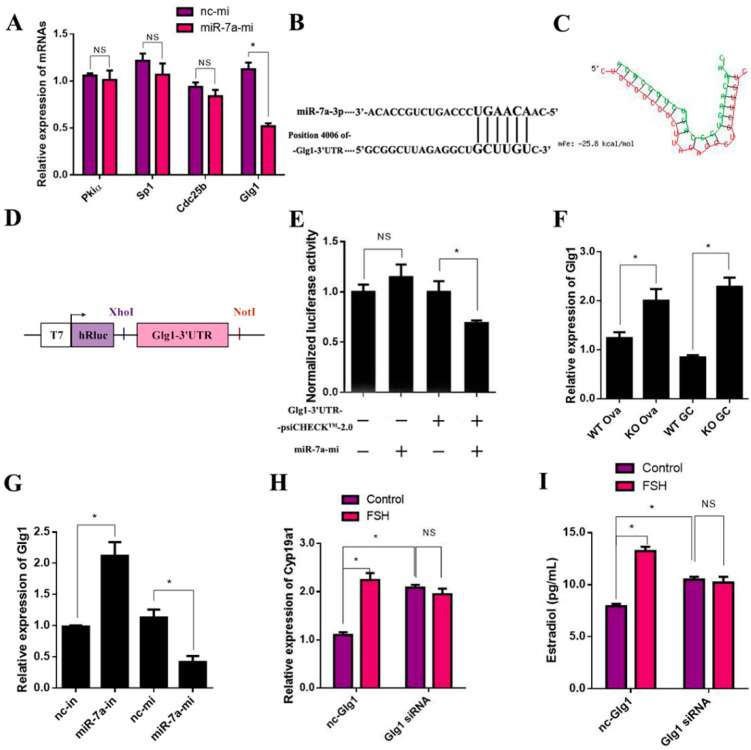
*G**lg1* is the direct target of miR-7a in ovarian granulosa cells. (**A**) Relative mRNA levels of *P**kiα*, *S**p1*, *C**dc25**b*, and *G**lg1* were detected by RT-qPCR and normalized to *GAPDH*; *n* = 3. (**B**) The predicted binding sites of miR-7a and 3′-UTR of *G**lg1*. (**C**) Schematic representation of the hybridization between miR-7 and Glg1 3′-UTR and the predicted minimum free energy, as calculated by RNA hybrid. Green letters indicate miR-7a sequences, and red letters indicate Glg1 sequences. (**D**) Schematic of the inserted GLG1 3′-UTR sequences into the psiCHECK^TM^-2.0 vector. (**E**) The relative luminescence intensity detected using a dual-luciferase reporter assay kit after the co-transfection of miR-7a-mi or nc-mi and the dual-luciferase vectors into 293FT cells; *n* = 3. (**F**) Relative *G**lg1* mRNA levels were detected by RT-qPCR and normalized to *GAPDH*; *n* = 3. (**G**) Relative *G**lg1* mRNA levels were detected by RT-qPCR and normalized to *GAPDH*; *n* = 3. (**H**) *C**yp19a1* mRNA levels in cells were analyzed by RT-qPCR and normalized to *GAPDH*; *n* = 3. (**I**) E_2_ levels in the culture medium were examined by RIA; *n* = 3. Data are presented as mean ± SEM. * *p* < 0.05. NS, not significant.

**Table 1 ijms-23-08565-t001:** PCR primer sequences.

Primer Names	Sequence
U6 RT Primer	AACGCTTCACGAATTTGCGT
U6 Forward Primer	CTCGCTTCGGCAGCACA
U6 Reverse Primer	AACGCTTCACGAATTTGCGT
miR-7a RT Primer	CTCAACTGGTGTCGTGGAGTCGGCAATTCAGTTGAGACAACAAAAT
miR-7a Forward Primer	GATGAGCTGTCCACCTGCTT
miR-7a Reverse Primer	CTGTCCCCTGTCCCACTCTA
GAPDH Forward Primer	GGTTGTCTCCTGCGACTTCA
GAPDH Reverse Primer	GGGTGGTCCAGGGTTTCTTA
Cyp19a1 Forward Primer	TTGGAAATGCTGAACCCCAT
Cyp19a1 Reverse Primer	CAAGAATCTGCCATGGGAAA
Glg1 Forward Primer	GGGCTGTACCTGACCTCT
Glg1 Reverse Primer	CCTTGTCACCACCTGTCT
17βHSD Forward Primer	GACCGTTCCCAGAGCTTCAA
17βHSD Reverse Primer	CAGCACCCACAGCGTTCAAT
Cyp17a1 Forward Primer	GATCTAAGAAGCGCTCAGGCA
Cyp17a1 Reverse Primer	GGGCACTGCATCACGATAAA

## Data Availability

The datasets generated during and/or analysed during the current study are available from the corresponding author on reasonable request.

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
