# Peer review of "MicroRNA-7a2 Contributes to Estrogen Synthesis and Is Modulated by FSH via the JNK Signaling Pathway in Ovarian Granulosa Cells"

_ijms, 2022, doi:10.3390/ijms23158565_

Round 1

Reviewer 1 Report

This manuscript “MicroRNA-7a2 contributes to estrogen synthesis and is modu- 2 lated by FSH via the JNK signaling pathway in ovarian granu- 3 losa cells “is interesting study for functions of MIRNA-7a2. However, authors should document more visible data and revise the manuscript with my comments.

Major comments

1] In the fig1, why did authors add FISH (image) results? For associating between them (a and c panels), authors should show double labeling results with markers of oocyte and granulosa cells. Additionally, authors should document experimental procedure for negative control   

2] Already, many studies documented functions of the miRNA in ovary hypogonadism and infertility by upregulation of FSH and LH. Although authors focused JNK signaling and the miRNA, results associated with the signaling was not enough.

3] Authors should show evidences for miRNA KO ovary using RT-PCR or FISH         

4] In Fig2, authors should show histological results for the panel F

Reviewer 2 Report

The study by Li et al., is well-designed and clearly described. It reports miRNA 7a2 role for granulosa cells in mouse ovary which is novel information.

Give full name for JNK, PKC, PKA, ERK etc.

 Lines 56-60-more detailed information should be provided

Figures should be larger, sometimes are too small

Round 2

Reviewer 1 Report

Congratulation !